# Device Orientation Independent Human Activity Recognition Model for Patient Monitoring Based on Triaxial Acceleration

Sara Caramaschi [1,2,*,†], Gabriele B. Papini [3,4,†] and Enrico G. Caiani [1,5,†]

1 Department of Electronics, Information and Bioengineering, Politecnico di Milano, 20133 Milan, Italy
2 Department of Computer Science and Media Technology, Internet of Things and People, Malmö University, 211 19 Malmö, Sweden
3 Department of Patient Care & Monitoring, Philips Research, 5656 AE Eindhoven, The Netherlands
4 Department of Electrical Engineering, Eindhoven University of Technology, 5612 AZ Eindhoven, The Netherlands
5 Istituto Auxologico Italiano, IRCCS, S. Luca Hospital, 20149 Milan, Italy
* Correspondence: sara1.caramaschi@mail.polimi.it
† These authors contributed equally to this work.

**Abstract:** Tracking a person's activities is relevant in a variety of contexts, from health and group-specific assessments, such as elderly care, to fitness tracking and human–computer interaction. In a clinical context, sensor-based activity tracking could help monitor patients' progress or deterioration during their hospitalization time. However, during routine hospital care, devices could face displacements in their position and orientation caused by incorrect device application, patients' physical peculiarities, or patients' day-to-day free movement. These aspects can significantly reduce algorithms' performances. In this work, we investigated how shifts in orientation could impact Human Activity Recognition (HAR) classification. To reach this purpose, we propose an HAR model based on a single three-axis accelerometer that can be located anywhere on the participant's trunk, capable of recognizing activities from multiple movement patterns, and, thanks to data augmentation, can deal with device displacement. Developed models were trained and validated using acceleration measurements acquired in fifteen participants, and tested on twenty-four participants, of which twenty were from a different study protocol for external validation. The obtained results highlight the impact of changes in device orientation on a HAR algorithm and the potential of simple wearable sensor data augmentation for tackling this challenge. When applying small rotations (<20 degrees), the error of the baseline non-augmented model steeply increased. On the contrary, even when considering rotations ranging from 0 to 180 along the frontal axis, our model reached a f1-score of $0.85 \pm 0.11$ against a baseline model f1-score equal to $0.49 \pm 0.12$.

**Keywords:** device displacement; acceleration; wearable devices; data augmentation; patient monitoring; human activity recognition

## 1. Introduction

The goal of Human Activity Recognition (HAR) is to classify the movement of a person into a pre-defined activity set. This information is used in multiple contexts, ranging from fitness tracking, health assessment and elderly care, to human–robot interaction [1–4]. In a clinical context, HAR can be used to outline the patients activities during hospitalization to improve, or enable, recovery/deterioration monitoring [5]; additionally, HAR allows for the contextualization of electrocardiographic patterns [6,7]. As the study from Brown et al. [8] stated, a low amount of dynamic activities may cause negative consequences in hospitalized patients; therefore, it appears relevant to recognize and quantify their activities to timely assist them. Wearable sensor technologies can support clinicians by providing tools for continuous measurement acquisition; however, the positioning of these devices is critical.

Once worn or applied to the body, sensor displacement can occur caused by wrong positioning, or by physical peculiarities of the patient such as gender, height, weight, and age [9]. Additionally, within a single position and if free of moving, the wearable device could have continuous unexpected movements, mostly of small entity [10]. HAR algorithms are mainly data-driven, meaning that poor results are expected upon random movements.

Most HAR approaches rely on machine learning (ML) techniques based on feature-based models or raw-data models. Among the commonly applied algorithms, it is possible to find Hidden Markov Models, Support Vector Machines (SVM), k-Nearest Neighbor, and Random Forest. In addition to ML techniques, Deep Neural Networks (DNN)-based algorithms are also used in the HAR context, considering a trade-off between model simplicity and interpretability, as mentioned by the review from Zhang et al. [11].

In a previous work on HAR from our team, Fridriksdottir et al. [12] described a DNN based on three-axial accelerometer to recognize hospitalized patients' activity and compared its results with those obtained with a feature-based SVM algorithm, where the best performance was achieved by the DNN approach (accuracy of 94.5% and f1-score of 94.6% against 83.35% and 85.07%, respectively, with SVM). This study was based on a single device, body-taped to the patient's chest to avoid its displacement. This way, the body positions range that could have been recognized was narrowed down to a limited set.

Other researchers used a combination of an initial position classifier with a subsequent position-dependent HAR algorithm. This concept relies on the assumption that HAR can be obtained from sensors applied on different body locations: Saeedi et al. [13] considered seven different locations (i.e., both ankles, wrists, left thigh, right arm, and waist), while Sztyler et al. [14] included the chest, forearm, head, shin, thigh, upper arm, and waist. In [10], different correction methods for sensor displacement were proposed, including both a feature-based approach and an ML classifier in an attempt to make the HAR model position independent. The proposed method was developed for a multi-sensor scenario.

As these studies showed, it is challenging to obtain representative data from multiple positions and their possible displacements. For example, pending devices (pendants) make the task difficult because of the countless movement combinations that could occur. In this context, we hypothesized that data augmentation techniques could help in synthesizing different sensors' configurations to artificially explore a wide range of possible scenarios.

Nowadays, data augmentation is standard practice when dealing with ML applied to images, to obtain additional information for the ML models and to avoid overfitting [15]. Typically applied image-augmentation techniques include geometric transformations, filtering, mixing images, random erasing, and feature space augmentation [16]. Wearable sensor data augmentation represents a less common approach field; however, it was shown to positively affect time-series based computation and to provide potential improvements in data-driven tasks such as HAR. The review from Zhang et al. [11] states that high quality data augmentation techniques are necessary for the growth of HAR research. Augmentation of wearable sensors data was firstly addressed by Ohashi et al. [17], proposing an augmentation strategy that considers the physical constraints of the arms applied to a multi-sensor scenario, including an accelerometer, a gyroscope, and an electromyography sensor. Steven et al. [18] proposed an ensemble data augmentation to the spectral feature space to improve activity recognition performances among only three classes (sitting, standing, and walking), reaching an accuracy of 88.87%. The study by Um et al. [19] proposed a method to classify the motor state of Parkinson's Disease patients by using data augmentation, where applying measurements rotation improved the performance compared to other techniques. Wang et al. [20] stated how HAR sensor data annotation represents a challenging task. To tackle it, they applied resampling augmentation of accelerometer data within a contrastive learning framework. This newly proposed approach learns representations by contrasting positive pairs, corresponding to the same sample augmentations, against negative pairs, or unrelated separated samples, helpful when few training data are available [21,22].

Device orientation is an important determinant when a three-axial acceleration solution for HAR is considered. Accordingly, the aim of this research was to investigate the

impact of changes in sensor orientation on a deep-learning (DL) HAR algorithm targeted on patient-like activities, such as slow and aided walking and wheelchair. Ultimately, we propose an orientation-independent HAR model that leverages data augmentation, and that is trained with acceleration measurements recorded from five sensor locations on the participant's trunk.

## 2. Materials and Methods

### 2.1. Dataset

Two datasets were considered in this research. The first was represented by the Wearing Position Study (WPS) acquired within Philips Research laboratories (2022). It contains three-axis acceleration measurements from nineteen healthy volunteers, ten males and nine females, while the second is the Simulated Hospital Study (SHS) acquired within Philips Research laboratories (2019). The SHS includes ten male and ten female healthy volunteers. Table 1 shows the age, weight, height, and BMI median and first and third quartiles of both WPS and SHS participants. Before starting the test, each participant was explained the protocol and afterwards was asked to signed an informed consent, obtained from all participants involved. Both studies, according to the regulations in the Netherlands, were waived as non-medical research, and therefore, approval by an IRB institution was not needed. The Internal Committee for Biomedical Experiments at Philips approved both studies. Each study was characterized by a specific protocol of activities to be followed (see Table 2) by the participants. The protocol was performed under the guidance and observation of two researchers, who annotated the start and end time for each activity. Self-paced activities (i.e., self-paced walking and self-push wheelchair) were acquired along a 30-meter corridor without obstacles.

In the WPS study, five GENEActiv (GA) accelerometers [23] were used. Two were applied on the skin of the participant on the left lower rib (GA lower rib) and on the chest (GA chest) using body tape, while the other three were applied on a rigid support that simulates the position of a patient monitor device, with two of them pending from the neck (GA front and GA side) and the third one placed inside the pocket of a clinical gown (GA gown). In the SHS study, the sensors' setting included the GA front only. Figure 1 shows examples of a patient monitoring device usage in two different positions, front and side. The sampling frequency of all accelerometers was set to 100 Hz with a dynamic range of $\pm 8$ g ($1g = 9.8 \, \text{m/s}^2$).

Once data acquisition was completed, signals were synchronized to the annotations based on the performed activities and synchronization patterns (i.e., three jumps at the beginning and end of the session). Signals were down-sampled to 16 Hz and split into windows of 6 s, with 4.5 s of overlap. No other preprocessing operation was applied; the development and testing of the models used raw acceleration data as input.

**Table 1.** Median, first (Q1) and third (Q3) quartiles of the Wearing Position Study (WPS, left side) and Simulated Hospital Study population (SHS, right side) characteristics: age, weight, height, and Body Mass Index (BMI).

| | Age | Weight [kg] | Height [cm] | BMI [kg/m²] | Age | Weight [kg] | Height [cm] | BMI [kg/m²] |
|---|---|---|---|---|---|---|---|---|
| Median | 41.5 | 71.5 | 174.5 | 23.05 | 44.5 | 75.0 | 175.0 | 25.34 |
| Q1 | 25.8 | 61.2 | 167.8 | 21.32 | 32.8 | 68.5 | 166.5 | 23.77 |
| Q3 | 53.3 | 79.5 | 184.8 | 25.12 | 54.3 | 86.5 | 182.0 | 26.28 |

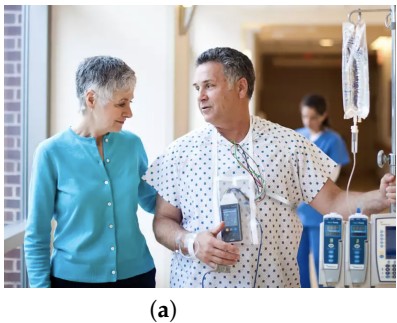 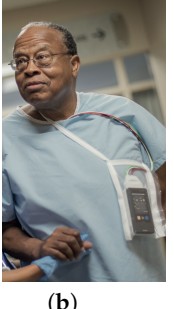 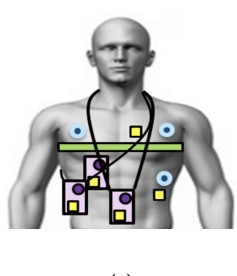

(**a**) (**b**) (**c**)

**Figure 1.** Patient monitoring device placement in the front (**a**) and side (**b**) positions [24,25]. (**c**) The sensor settings for the Wearing Position Study data collection. Yellow squares are GENEActive sensors: two in contact with the skin, two pending from the neck and one in the clinical gown pocket.

**Table 2.** Wearing Position Study and Simulated Hospital Study activities reported in chronological order.

| Activity | Duration | Activity | Duration |
|---|---|---|---|
| Jump 3x (sync) ** | Self-paced | | |
| Lying in bed ** | 3 min | Physioterapy on chair ** | 2 min |
| Left side ** | 30 s | Patient transport in wheelchair ** | 1 min |
| Right side ** | 30 s | Wheelchair self-push | Self-paced |
| Reclined ** | 30 s | Crutches ** | Self-paced |
| Upright ** | 30 s | Anterior walker ** | Self-paced |
| Sitting edge of the bed ** | 30 s | IV pole ** | Self-paced |
| Standing ** | 30 s | 4-wheel rollator ** | Self-paced |
| 0.6 km/h ** | 2 min | Walk slow * | Self-paced |
| 0.8 km/h ** | 2 min | Walk normal * | Self-paced |
| 1.0 km/h ** | 2 min | Walk fast * | Self-paced |
| 1.5 km/h ** | 2 min | Intermittent walking * | Self-paced |
| 2.0 km/h ** | 2 min | Shuffling * | Self-paced |
| 3.0 km/h ** | 2 min | Upstairs one leg first ** | Self-paced |
| 4.0 km/h ** | 2 min | Downstairs one leg first ** | Self-paced |
| 4.0 km/h inclined * | 2 min | Stairs ascent ** | Self-paced |
| Washing hands ** | 1 min | Stairs descent ** | Self-paced |
| Reading ** | 1 min | Jump 3x (sync) ** | Self-paced |

*: Activities performed only in the Wearing Position Study (WPS); **: Activities performed in the Wearing Position Study and in the Simulated Hospital Study (SHS).

## 2.2. Model Architecture

The implemented HAR model architecture is shown in Figure 2 and represents a modified version of the DNN proposed by Fridriksdottir et al. [12]. The main difference with the previous model consists of the substitution of the Long Short Time Memory layer with a convolutional layer: this change in architecture was introduced to simplify the model and it did not generate results significantly different from the previous DNN. The model input consists of the X-, Y-, and Z- acceleration segments of shape *(number of segments, 96, 3)*. The model includes four 1D convolutional layers interspersed with four batch normalization layers. Moreover, two max-pooling and dropout layers were added to reduce overfitting risks. After the fourth batch normalization layer, a flattening layer was added to reshape the data and to provide input for the final dense layer that computes the prediction probabilities of five classes, by means of a softmax activation function [26].

The model uses categorical cross entropy as loss function, and the 'Adam' optimizer [27], considering a batch size of 100 samples. The 'Balanced Batch Generator' function was used to fit the model: it is a Keras [28] function that allows creating balanced batches during model training by specifying the desired sampler, where in this case a random sampler was applied.

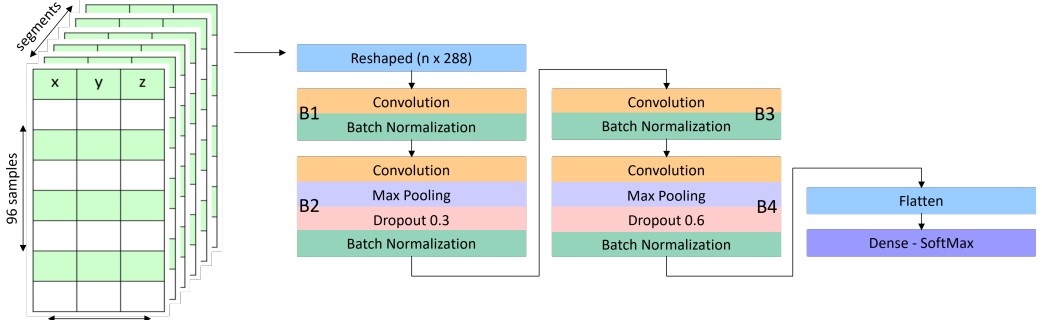

**Figure 2.** Convolutional Neural Network model architecture composed of an input layer, four blocks including multiple layers (B1, B2, B3, B4), flattening and ending softmax layer.

### 2.3. Data Augmentation

Data augmentation was used to synthesize different points of view relevant to the same data [29]. A rotational matrix was applied to the original acceleration measurements as in the Equation:

$$\begin{bmatrix} acc'_x \\ acc'_y \\ acc'_z \end{bmatrix} = R_x R_y R_z \begin{bmatrix} acc_x \\ acc_y \\ acc_z \end{bmatrix} \tag{1}$$

In particular, $(acc_x, acc_y, acc_z)$ are the original values and $(acc'_x, acc'_y, acc'_z)$ are the computed acceleration values associated with the applied rotation. The chosen rotation angle is $\alpha$, in degrees units. The rotational matrixes are defined for each axis and correspond to $R_x, R_y, R_z$:

$$R_x = \begin{bmatrix} 1 & 0 & 0 \\ 0 & \cos(\alpha) & -\sin(\alpha) \\ 0 & \sin(\alpha) & \cos(\alpha) \end{bmatrix} \tag{2}$$

$$R_y = \begin{bmatrix} \cos(\alpha) & 0 & \sin(\alpha) \\ 0 & 1 & 0 \\ -\sin(\alpha) & 0 & \cos(\alpha) \end{bmatrix} \tag{3}$$

$$R_z = \begin{bmatrix} \cos(\alpha) & -\sin(\alpha) & 0 \\ \sin(\alpha) & \cos(\alpha) & 0 \\ 0 & 0 & 1 \end{bmatrix} \tag{4}$$

### 2.3.1. Augmentation Setting for Training Data

The number and range of rotations applied to the accelerometer signals might affect the success of data augmentation. Therefore, we initially tested which rotation pattern resulted in the largest performance improvement for our model during cross-validation. Two augmentation training datasets were considered: the first set consisted of seven rotations between 0 and 90 degrees, while the second set consisted of seven rotations between 0 and 180 degrees. Rotations were applied separately along the frontal and sagittal axis of the human body. The frontal axis splits the body into a dorsal and ventral parts, while the sagittal axis splits the body into an upper and lower halves. To compare the two augmented sets, tests were made for rotations from 0 to 360 degrees with a step of 5 degrees.

Based on this preliminary analysis, the final augmentation settings for the training set of the augmented model consisted of ten rotations from 0 to 180, with a 20 degree step on the frontal and sagittal axis separately. While all the original acceleration signals were considered in the training data, only a randomly selected portion (11.1%) of these signals was kept when applying a single rotation. The baseline model was trained using the data collected by the five upper body sensors in the WPS dataset, including a total of 73,683 segments (windows of 6 s overlapped by 4.5 s). The resulting training size of the augmented model was three times the size of the baseline training set.

2.3.2. Augmentation Setting of Testing Data

The models were evaluated through three different test sets shown in Figure 3 and reported below:

- Original: this test set did not have any data augmentation.
- Real-life test set: double-axis small rotations along the frontal and sagittal axis (respectively X- and Z-axis). In particular: [[5, 5], [5, 2], [2, 5], [10, 10], [10, 5], [5, 10], [15, 15], [15, 10], [10, 15]], unit of measurement in degrees.
- Fully-rotated test set: fifty-six rotations between 0 and 360 degrees applied along the frontal, longitudinal, and sagittal axis (respectively, X-,Y-, and Z-axis) separately.

Table 3 describes the selected participants and sensors used for training and testing of both the baseline and augmented HAR models. Fifteen participants of the WPS were considered when cross-validating the model: ten for training, two for validation, and three for testing. The participants in the cross-validation procedure were randomly split and performance for each fold was observed to see if there were any discrepancies between the splits. On the other hand, four random participants of the WPS and all twenty participants of the SHS were kept separated and considered only in the final testing as a holdout set (i.e., external validation).

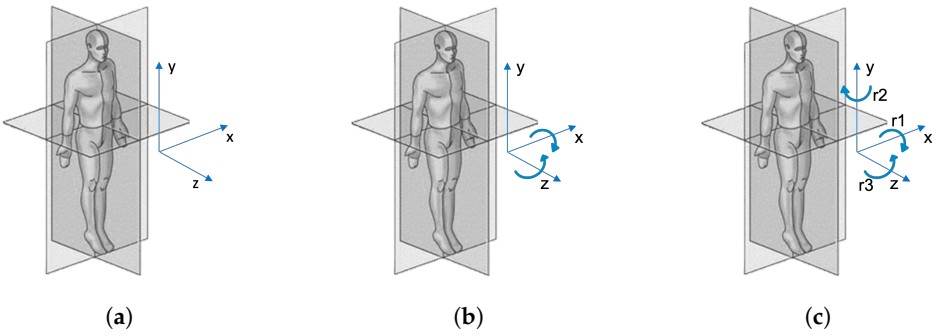

(**a**)          (**b**)          (**c**)

**Figure 3.** Augmented test set visualization. (**a**) The original sensor orientation compared to the standing human body. (**b**) The applied rotations of the real-life test set along the frontal and sagittal axis. (**c**) The three non-simultaneous rotations applied along the frontal, longitudinal, and sagittal axis (r1, r2, r3) with the fully-rotated test set.

**Table 3.** Train and test settings for the baseline and the augmented model computation. The baseline model training did not undergo data augmentation. The two models were tested in the same way by means of three test sets.

| Train/Test | Participants | Rotations | Rotation Axis | Sensors' Location |
|---|---|---|---|---|
| Train baseline model | WPS—15 participants | - | - | Front, side, gown, chest, left lower rib |
| Train augmented model | WPS—15 participants | 0 to 180 deg. step 20 | Frontal (X-axis), sagittal (Z-axis) | Front, side, gown, chest, left lower rib |
| Test holdout | WPS—4 holdout participants | Test sets: Original, real-life, fully-rotated test sets | | Front, side, gown |
| | SHS—20 participants | | | Front |

## 3. Evaluation of the Orientation Impact Model and HAR Performance

A five-fold cross-validation [30] was used to train both the baseline and the augmented models. The cross-validation performance was used to determine the augmentation approach (i.e., the range of rotations), and the effect of the rotation on the baseline model.

Each cross-validation fold used the WPS data of ten participants for training the model, the data of two participants for early stopping, and the data of three participants to assess the model performance.

The activity labels of the two holdout sets were estimated by a majority-voting ensemble of the results of the five models obtained during cross-validation for baseline and augmented models. The results of the original test sets were averaged over the considered participants. The real-life and fully-rotated test sets' results were averaged on the applied external rotations and the considered f1-score was computed by micro-averaging obtained predictions. The performance metrics that evaluated each class were the f1-score, precision, recall, and specificity. Additionally, the Cohen's Kappa (Kappa-score) was considered: it represents an inter-rater agreement coefficient between two raters, as a function of the probability that the two raters are in perfect agreement [31]. For statistical analysis, first the Shapiro–Wilk [32] test was used to verify the normality of f1-score values. Then, the Wilcoxon signed-rank and the t-test were applied to establish differences within performance distributions. The Wilcoxon test is non-parametric, and therefore, does not require normality of the observed data [33].

## 4. Results

### 4.1. Rotation Impact on the Baseline Model

The baseline model was tested by using data augmentation, in particular, by applying rotations, from 0 to 180 degrees, on the frontal, longitudinal, and sagittal axis. Performances between the five cross-validation splits were observed. In particular, a minimum Kappa-score value of 0.87 and a maximal Kappa-score value of 0.92 were obtained when testing the baseline model with the original test set. Thus, it could be concluded that the model performance was not dependent on which recordings were included in the training set. Figure 4 reports the percentage of wrong classifications according to multiple axis and groups of activities. It is noticeable how the Y-axis, parallel to the participant's frontal plane and parallel to the Earth's gravity acceleration, was the least impacted axis by orientation changes. Moreover, because of the nature of the proposed activities, the static ones (Lying in bed, Left side, Passive wheelchair, Right side, Reading, Reclined, Sitting on the edge of the bed, Standing, Upright, Washing hands/brushing teeth) were the least affected compared to the dynamic ones.

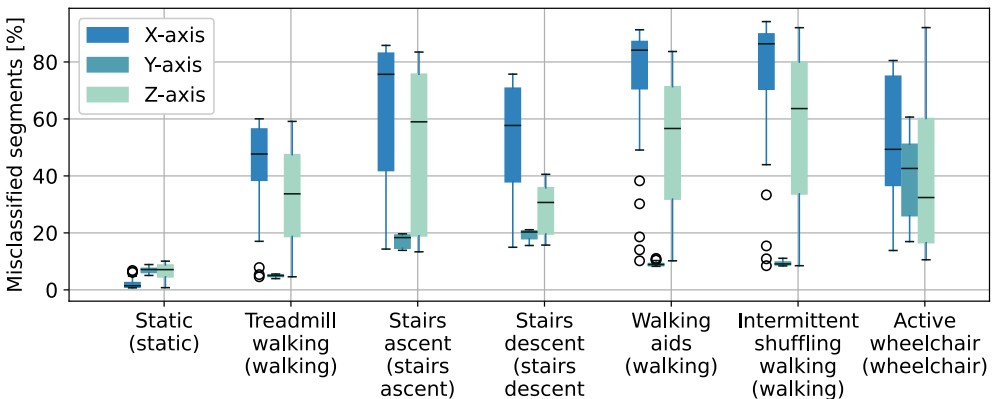

**Figure 4.** Percentage of misclassified segments when testing the baseline model with rotations from 0 to 180 degrees applied to each axis separately. Misclassified segments correspond to the amount of false negative predictions for the specific label

Figure 5 shows the percentage of misclassified segments of selected groups of activities when rotations were applied along the frontal, longitudinal, and sagittal axis (respectively, X-, Y-, and Z-axis) separately; the performance is shown for each applied rotation from 0 to 180 degrees (0, 5, 10, 20, 25, 30, 40, . . . , 180 degrees). Two groups of activities are reported: treadmill walking and static activities. As previously observed, major differences were

noticeable in relation to the type of activity being considered and the axis to which the rotation was applied. It is relevant to highlight that, within dynamic activities, the error percentage increased rapidly even for low values of applied rotations. Activity distribution from the WPS and SHS data, divided according to the label of our interest, were as follows: stairs ascent 5.2%, stairs descent 4.9%, static 25.2%, walking 62.5%, wheelchair 2.3%.

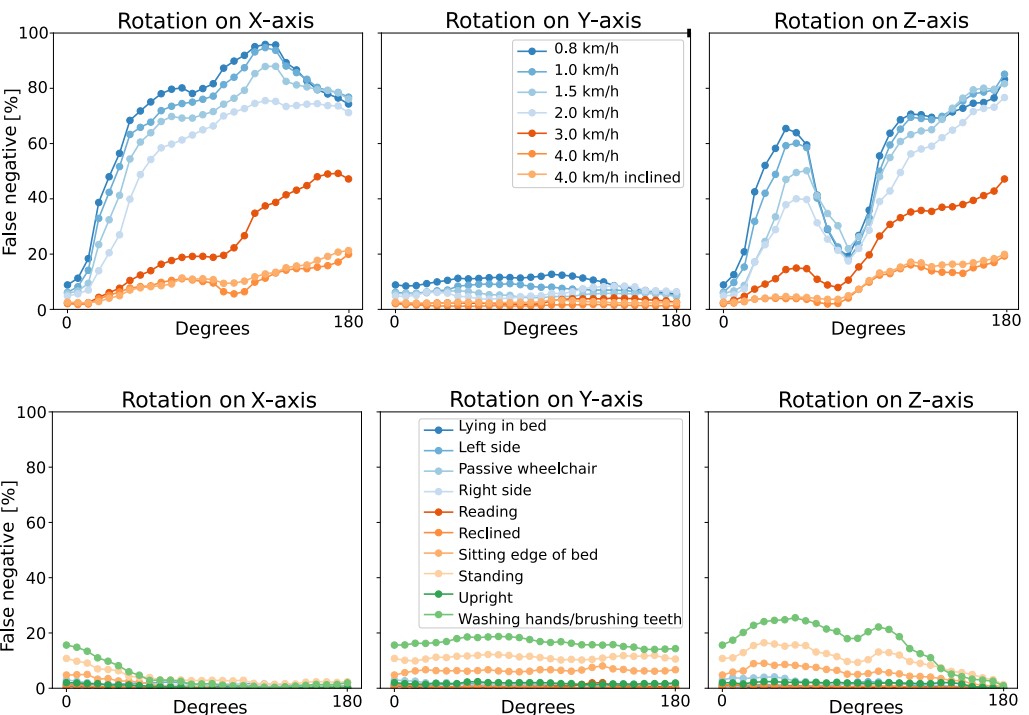

**Figure 5.** Top panels show false negative percentage profiles of treadmill walking activities; bottom panels report false negative percentage profiles of static activities. Rotations on the X-, Y-, and Z-axis correspond to rotations applied along the frontal, longitudinal, and sagittal axis, respectively.

### 4.2. Augmentation Approach

The performance of models trained with two augmented training sets was observed to determine which one would suit best. In particular, the comparison between the two ranges of rotation was computed from 0 to 360 degrees, every five degrees, and results were presented for each 90 degree range. As shown in Figure 6, the model trained with a range of rotations that span from 0 to 180 had better results over three quarters out of four, in terms of Kappa score.

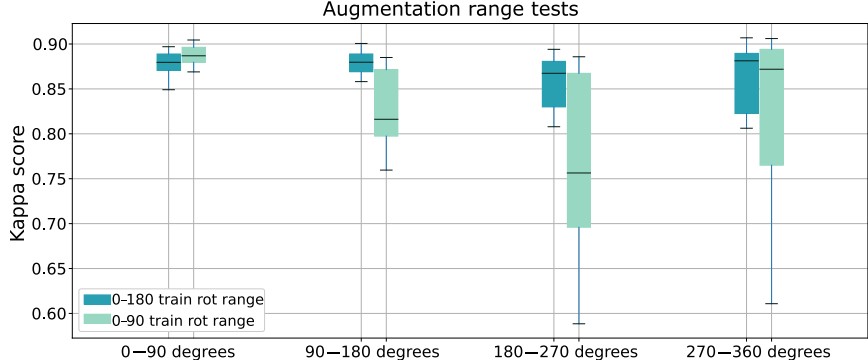

**Figure 6.** Performances comparison of Kappa score between two differently augmented models.

### 4.3. Holdout Data Results—External Validation

When rotations were applied on a single or double axis, the baseline model significantly increased the errors when classifying what activity the participant was doing, thus decreasing the performance. The augmented model maintained high performance even when rotations were applied. This consideration was confirmed from testing outcomes on both WPS holdout and SHS data, as external validation. Figure 7 reports the obtained results for the baseline and the augmented model according to the three augmented test sets in terms of f1-score and Kappa-score. Appendix A reports detailed numerical results, such as the median and interquartile range referring to Figure 7 and single class results according to each model, each testing set, and considered participants.

Statistical analysis was conducted on the f1-score and Kappa-score of the two models. For all data (cross-validated participants, WPS holdout participants, and SHS external validation), no significant difference was observed related to the original test set. The paired t-test was applied to test real-life test set performances, obtaining a *p*-value smaller than 0.01 for both holdout source sets. The Wilcoxon-rank test was used for the fully-rotated test set, showing a *p*-value < 0.01 for each axis of the WPS holdout data. The SHS *p*-values were below 0.01 for both the frontal and sagittal axis (X- and Z-axis), while for rotations applied along the longitudinal axis (Y-axis), no significant difference was shown.

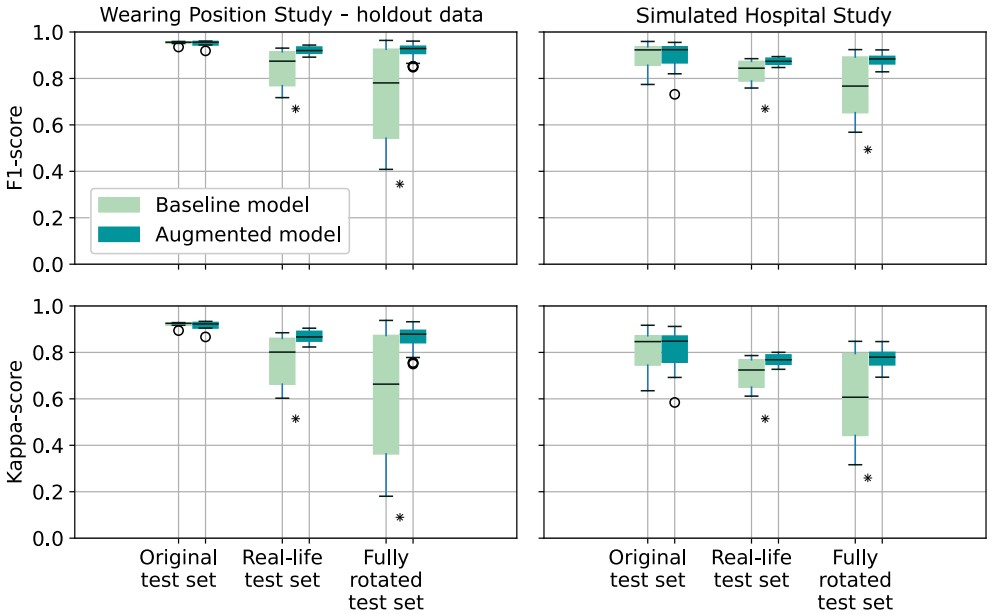

**Figure 7.** F1-score (top panels) and Kappa-score (bottom panels) of the baseline and the augmented model for the three test sets: Original, real-life and fully rotated test sets. *: test sets obtained statistically different results for the baseline and augmented model with a *p*-value < 0.01.

Additionally, Figure 8 highlights the covered area of false negative percentage profile related to treadmill walking activities. The green area belongs to the baseline model and it is generally wider than the one of the augmented model. Low-speed treadmill activities (⩽2.0 km/h) majorly contributed to the upper part of the green area. On the contrary, high-speed treadmill activities (>2.0 km/h) had generally fewer false negatives (lower part of the green area). This behavior was less visible in the profiles of the augmented model.

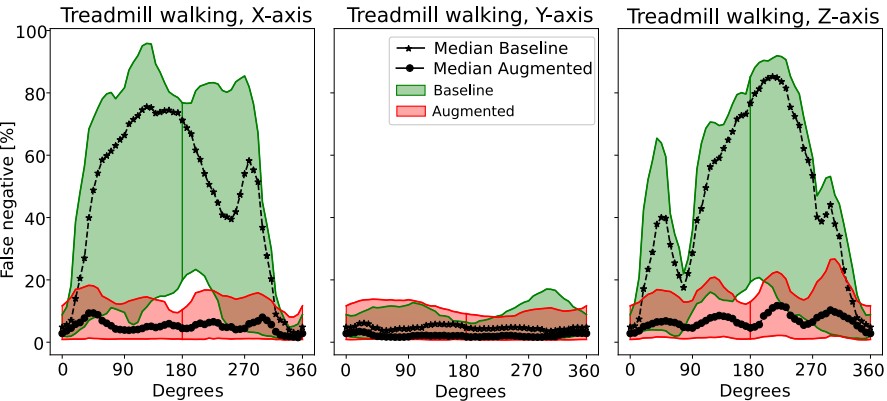

**Figure 8.** False negative percentage profiles of treadmill activities (i.e., walking class) for rotations applied to each axis separately for baseline and augmented models. The top line corresponds to the maximum profile for that rotation; the bottom line corresponds to the minimum profile for that rotation; the black dotted lines correspond to the profile median value.

## 5. Discussion

The context of HAR is broad and with multiple fields of application, making it hard, sometimes, to compare studies due to the diversity in the selected activities, environment conditions, target population, and chosen metrics [34].

Our study was focused on simulated activities that may characterize a hospitalized patient wearing a device with freedom of movement (i.e., pendants or inside a pocket), localized in the upper part of the patient's body. Thanks to data augmentation, the HAR model was able to learn additional configurations not provided by the initial dataset. As Figure 8 shows, the false negative percentage red area covered by the augmented model was significantly smaller compared to the green area belonging to the baseline model. Additionally, the red area kept the error profile low and stable while the applied rotations increased.

To choose the augmented rotation ranges to be applied to the training data, performances obtained from two different sets, shown in Figure 6, were evaluated: despite the seven rotations between 0 and 180 degrees being sparse, they allowed the model to better learn device configurations characterized by higher applied rotations. In light of these considerations, a rotation range from 0 to 180 degrees was chosen for training data augmentation.

To the best of our knowledge, few research studies addressed data augmentation of acceleration signals; therefore, expanding this research field and its potential applications could be of relevant interest in this knowledge domain. The study by Ohashi et al. [17] addressed data augmentation according to a specific physical constraint; in particular, it allows sensor movement only on a certain trajectory dictated by the arm's degrees of freedom. In contrast, our applied augmentation does not follow physical constraints. In fact, it includes rotations that could easily happen when using sensors pending from the body in patient monitoring devices (i.e., the Portrait[TM] Mobile, by GE Healthcare [35], or the IntelliVue MX40 by Philips [36]), such as for example: the up-side down flipping of the device (i.e., 180 degrees on the frontal axis), inclined device due to body shape (i.e., small rotations along the frontal axis), and inclined device due to asymmetric position of the pending rigid support (small rotations along sagittal axis).

Collecting data spanning from many orientation configurations is highly time- and computationally expensive; from this perspective, data augmentation could represent an optimal approach to deal with this aspect and to increase overall performance. In Table 4, the main augmentation-related studies found in the literature are reported, along with the considered sensors and their positions, the applied augmentations, and the identified activities within the proposed framework.

**Table 4.** Description of the most relevant studies related to wearable sensor data augmentation in the context of HAR. Each study is described by its sensors' positioning, applied augmentation, type of sensor, and recognized activities.

| Author | Sensors' Position | Applied Augmentation | Augmented Sensors | Recognized Activities |
|---|---|---|---|---|
| [17] | Forearm | Rotations around X-axis | Accelerometer Gyroscope EMG * | Holding, Twisting, Folding |
| [18] | Left wrist | Averaging, combining, shuffling | Spectral features of accelerometer and gyroscope | Sitting, Standing, Walking |
| [19] | Wrist | Rotation, Permutation, Time-warping, Magnitude-warping | Accelerometer | Motor state of Parkinson sbj: Bradyiknesia, Dyskinesia |
| [20] | Mobile phone pockets | Resampling for contrastive learning | Accelerometer Gyroscope Magnetometer | UCI-HAR [37], MotionSense [38], USC-HAD [39] |
| Proposed model | Body trunk | Rotations around the three axis separately | Accelerometer | Stairs up, Stairs down, Static, Walking, Wheelchair |

*: augmented sensors that undergo different kind of augmentation than the one reported in the table.

In accordance with the literature, our initial results confirmed that device displacement might cause significant performance loss when using sensor orientation-dependent models. The error rate steeply rises even with small rotations (i.e., 5 degrees applied to the frontal axis ≈ 10%; 10 degrees applied to the frontal axis ≈ 20% for 0.8 km/h from Figure 5). False negative percentages of static activities did not increase when rotations were applied (i.e., 5 and 10 degrees applied to the frontal axis ≈ 5% for "Sitting edge of the bed" from Figure 5). As a result of the stable acceleration pattern, the model was able to recognize and classify this behavior as static activity. On the other hand, treadmill-related activity results showed an error rise as the applied rotation increased over the frontal and sagittal axis (X- and Z-axis). This trend was probably due to the nature of the different treadmill walking activities. In particular, high-speed walking activities had a low error profile. This activity type showed high peaks during the heel-strike and toe-off gait phases, allowing the model to predict it more easily. However, even for high-speed walking activities, their error percentage increased when larger rotations were applied (≈90 degree on the frontal axis). A possible reason for this could be that large rotations along the frontal and sagittal axis (X- and Z-axis) implied switching the acceleration component parallel to gravity that usually carries most part of the information. Figure 9 shows an example of walking activities of 0.8 km/h and 4.0 km/h and their corresponding applied rotations of twenty and sixty degrees. It was visible how slower walking had a smaller acceleration range. On the other hand, faster walking acceleration range had more dynamism and the information spanned a wider acceleration range.

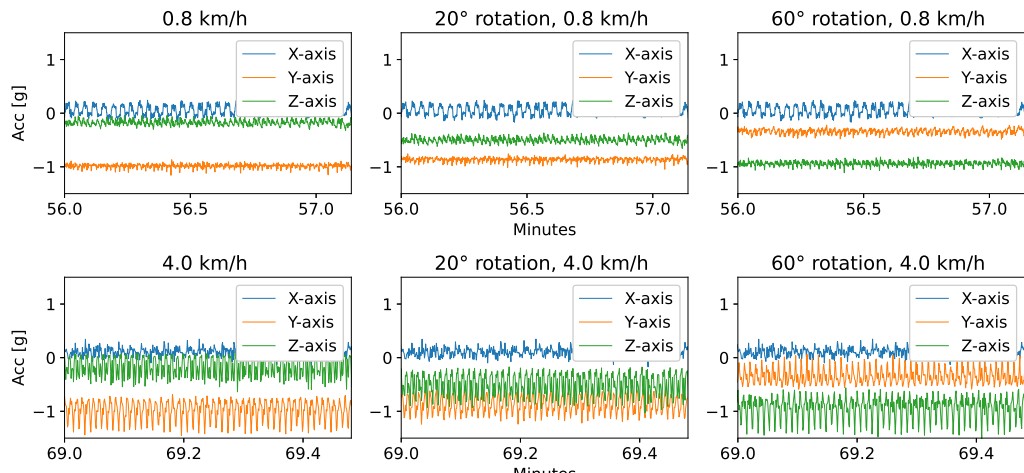

**Figure 9.** Walking activities of 0.8 km/h (top) and 4.0 km/h (bottom) and the corresponding applied rotations of twenty and sixty degrees ($g = 9.81$ km/s$^2$).

As shown in Figure 4, the static activities were the least affected by rotations on all orientation axes, probably due to their low acceleration values. Considering the models' performance differences among activities, another possible approach could be a tailored augmentation to the activity itself. Future work might consider transformations applied only to the classes that are majorly influenced by rotation, i.e., the classes with a high dynamic range, in terms, for example, of acceleration magnitude. This way, redundancy would be avoided and data would be augmented more efficiently.

*Limitations and Future Work*

Among the five prediction classes, further processing could be applied to the "wheelchair" class. As a matter of fact, it was easy to misclassify it with static or walking activities, due to acceleration pattern similarities. Our performance showed a low precision for the original test set and a high value for recall of the wheelchair class (0.41 ± 0.23 precision, 0.92 ± 0.12 recall for SHS participants, original test set). Frequently, slow walking activities were wrongly classified as wheelchair. A possible future improvement to "wheelchair" precision could be to apply post-processing steps to the predicted "wheelchair" class. For example, contextualizing the single "wheelchair" segment with the surrounding ones (i.e., within a certain number of consecutive "walking" segments, if "wheelchair" is detected, that prediction will likely be wrong, and therefore, post-processed as "walking"). Further steps could also consider prediction contextualization for all classes, either through post-processing or by adding specific layers to the deep-learning model (i.e., recurrent layer). Additionally, future studies should collect an higher amount of "wheelchair", "stairs ascent", and "stairs descent" data, given the imbalance of such data classes used in these studies.

Our application used accelerometer sensors; however, multiple studies have combined together different sensor modalities belonging to Inertial Measurement Unit technology, involving measurements of accelerometer, gyroscope, and sometimes, magnetometer data. Jiang et al. [40] proposed a method that merges accelerometer and gyroscope data into an activity image. They used CNN power and obtained outstanding accuracy performances related to three different public datasets. Using these sensors could be helpful, respectively, for different types of activities. For example, stairs ascending and descending performances could be improved using gyroscope or barometer data, while more dynamic activities, such as walking, rely on acceleration data. In most circumstances, acceleration measurements primary lead the activity classification, while gyroscope data have a secondary support role [41]. In spite of the fact that more signals and sensors could be integrated, we focused our research on a single triaxial accelerometer-based solution. This approach has the

advantage of being easily applicable to any device that contains an accelerometer, without the need to re-design its components. Additionally, it maintains low power consumption.

The used model architecture, CNN, lies within the most common DNN-based approaches [11]. Despite many advantages and progresses made through DNN-based models, multiple challenges still apply to these techniques, such as their explainability and generalization capabilities compared with models built on extracted features from the respective knowledge domain [42]. Considering the SHS study, within the context of a patient-monitoring solution, using DL capabilities proved to be effective and promising [12]. Through this study, we examined how rotations could impact DL algorithms and how data augmentation address for this aspect. This challenge might be found also for other ML approaches that are orientation-dependent (e.g., orientation dependent features, three-axis acceleration). Future work should focus on comparing the augmented DL approach with other techniques, such as HMM, or feature-based models [43,44].

The good performances of the augmented model obtained during cross-validation were confirmed by the holdout data results. This indicates that our model can well generalize using unseen data, i.e., participants. However, holdout data belong to the same study (WPS) or to a similarly acquired one (SHS) compared to training data, and while the participants were different between the sets, the activities performed were similar. This might have partially biased the performance of the classification algorithms that still needs to be confirmed in a real-life scenario. Despite this, the SHS was a different research study compared to the WPS and added additional holdout data. Moreover, many activities of the protocols were self-paced, meaning that each participant could choose their own walking speed (i.e., slow, fast, normal walk, and the aided-walking activities), and thus, adding data variability. Studies that include acceleration measurements whose source is a clinical population would help better define the generalization capabilities of the model.

## 6. Conclusions

This research investigated the effects of device displacement on a DNN-based HAR model performance and proposed an orientation-independent HAR model. Further relevant steps might relate to model testing on a real clinical population and to wearable sensor data augmentation using other approaches, such as activity-tailored augmentation.

By applying HAR to wearable devices, it is possible to monitor and classify the activities performed by a patient. Device displacement is among the biggest challenges related to wearable sensors. A primary analysis showed how displacement, even of small entity, could negatively impact HAR algorithm performance. Ultimately, we developed an orientation-independent model that classified five pre-defined activities within a range of actions likely to happen in a clinical environment. Through this research, a possible solution was proposed for device displacement in HAR, and new challenges were highlighted to broaden this field and get closer to better activity monitoring solutions for clinicians and patients.

**Author Contributions:** Conceptualization, S.C., G.B.P. and E.G.C.; methodology, S.C., G.B.P. and E.G.C.; software, S.C. and G.B.P.; validation, S.C. and G.B.P.; formal analysis, S.C., G.B.P. and E.G.C.; investigation, S.C. and G.B.P.; resources, S.C. and G.B.P.; data curation, S.C. and G.B.P.; writing—original draft preparation, S.C.; writing—review and editing, S.C., G.B.P. and E.G.C.; visualization, S.C.; supervision, G.B.P. and E.G.C.; project administration, G.B.P. All authors have read and agreed to the published version of the manuscript.

**Funding:** This research received no external funding.

**Institutional Review Board Statement:** Ethical review and approval were waived for this study, due to its consideration as non-medical research, and therefore, approval by an IRB institution was not needed. The Internal Committee for Biomedical Experiments at Philips approved the collection of data used in this research.

**Informed Consent Statement:** Informed consent was obtained from all subjects involved in the study.

**Data Availability Statement:** Restrictions apply to the availability of these data.

**Acknowledgments:** We acknowledge Lieke Cox's support during data acquisition.

**Conflicts of Interest:** Gabriele B. Papini is a Philips employee.

**Appendix A**

Here, we report numerical performances related to the baseline and augmented model in particular, as referred to in Figure 4 (Table A1) and Figure 7 (Table A2) .

**Table A1.** Median and IQR of false negative percentages according to different activity groups.

|  | X-Axis | Y-Axis | Z-Axis |
|---|---|---|---|
| **Static (static)** | 1.43 (1.58) | 7.14 (1.45) | 7.11 (4.19) |
| **Treadmill walking (walking)** | 47.69 (18.02) | 5.01 (0.62) | 33.71 (28.74) |
| **Stairs ascent (stairs ascent)** | 75.67 (41.21) | 18.34 (4.6) | 58.99 (56.78) |
| **Stairs descent (stairs descent)** | 57.71 (32.85) | 20.32 (2.78) | 30.69 (16.24) |
| **Walking aids (walking)** | 84.13 (16.58) | 8.9 (0.75) | 56.65 (39.47) |
| **Intermittent shuffling(walking)** | 86.38 (19.42) | 9.09 (1.0) | 63.63 (46.32) |
| **Active wheelchair(wheelchair)** | 49.33 (38.33) | 42.62 (25.06) | 32.41 (43.6) |

**Table A2.** Median and IQR of f1-score and Kappa-score according to the two holdout sets and three test sets.

|  | Test Set | Mdl | Original | Real-Life | Fully-Rotated |
|---|---|---|---|---|---|
| **f1-Score** | **WPS Holdout** | **Base.** | 0.96 (0.01) | 0.80 (0.15) | 0.78 (0.38) |
|  |  | **Aug.** | 0.96 (0.02) | 0.92 (0.03) | 0.92 (0.03) |
|  | **SHS** | **Base.** | 0.92 (0.08) | 0.84 (0.09) | 0.77 (0.24) |
|  |  | **Aug.** | 0.92 (0.07) | 0.87 (0.03) | 0.88 (0.03) |
| **Kappa-score** | **WPS Holdout** | **Base.** | 0.93 (0.01) | 0.80 (0.20) | 0.66 (0.51) |
|  |  | **Aug.** | 0.92 (0.02) | 0.87 (0.04) | 0.88 (0.05) |
|  | **SHS** | **Base.** | 0.85 (0.13) | 0.72 (0.12) | 0.61 (0.35) |
|  |  | **Aug.** | 0.85 (0.11) | 0.77 (0.04) | 0.78 (0.05) |

Following the numerical results of the baseline and augmented models of holdout data, each of the three proposed test sets were divided in subsections.

*Appendix A.1. Original Test Set Results for Individual Classes*

Table A3 reports the results of the original test set referring to the WPS holdout participants. Table A4 reports the same results but referring to the SHS participants.

**Table A3.** Single class results of baseline and augmented model for the original test set. Considered data: WPS holdout participants.

| Mdl | Metric | Stairs Ascent | Stairs Descent | Static | Walking | Wheelchair |
|---|---|---|---|---|---|---|
| **Base.** | **precision** | 0.89 ± 0.03 | 0.99 ± 0.01 | 0.98 ± 0.01 | 0.97 ± 0.01 | 0.70 ± 0.13 |
|  | **recall** | 0.88 ± 0.02 | 0.86 ± 0.06 | 0.96 ± 0.03 | 0.97 ± 0.02 | 0.93 ± 0.06 |
|  | **f1-score** | 0.89 ± 0.02 | 0.92 ± 0.04 | 0.97 ± 0.01 | 0.97 ± 0.01 | 0.79 ± 0.1 |
|  | **specificity** | 0.99 ± 0.0 | 1.00 ± 0.0 | 0.99 ± 0.0 | 0.95 ± 0.02 | 0.99 ± 0.01 |
| **Aug.** | **precision** | 0.88 ± 0.09 | 0.94 ± 0.05 | 0.9 ± 0.07 | 0.97 ± 0.01 | 0.41 ± 0.23 |
|  | **recall** | 0.89 ± 0.09 | 0.88 ± 0.12 | 0.94 ± 0.04 | 0.89 ± 0.09 | 0.92 ± 0.12 |
|  | **f1-score** | 0.88 ± 0.06 | 0.90 ± 0.08 | 0.92 ± 0.03 | 0.92 ± 0.05 | 0.53 ± 0.23 |
|  | **specificity** | 1.00 ± 0.0 | 1.00 ± 0.0 | 0.99 ± 0.0 | 0.93 ± 0.02 | 0.98 ± 0.01 |

**Table A4.** Single class results of baseline and augmented model for the original test set. Considered data: SHS participants.

| Mdl | Metric | Stairs Ascent | Stairs Descent | Static | Walking | Wheelchair |
|---|---|---|---|---|---|---|
| **Base.** | **precision** | 0.83 ± 0.09 | 0.94 ± 0.05 | 0.92 ± 0.06 | 0.97 ± 0.01 | 0.39 ± 0.21 |
| | **recall** | 0.92 ± 0.07 | 0.88 ± 0.12 | 0.92 ± 0.04 | 0.89 ± 0.09 | 0.91 ± 0.15 |
| | **f1-score** | 0.87 ± 0.06 | 0.90 ± 0.07 | 0.92 ± 0.03 | 0.92 ± 0.05 | 0.51 ± 0.23 |
| | **specificity** | 0.99 ± 0.01 | 1.0 ± 0.0 | 0.97 ± 0.02 | 0.94 ± 0.02 | 0.95 ± 0.06 |
| **Aug.** | **precision** | 0.88 ± 0.09 | 0.94 ± 0.05 | 0.90 ± 0.07 | 0.97 ± 0.01 | 0.41 ± 0.23 |
| | **recall** | 0.89 ± 0.09 | 0.88 ± 0.12 | 0.94 ± 0.04 | 0.89 ± 0.09 | 0.92 ± 0.12 |
| | **f1-score** | 0.88 ± 0.06 | 0.90 ± 0.08 | 0.92 ± 0.03 | 0.92 ± 0.05 | 0.53 ± 0.23 |
| | **specificity** | 1.0 ± 0.01 | 1.0 ± 0.0 | 0.96 ± 0.03 | 0.94 ± 0.03 | 0.95 ± 0.06 |

*Appendix A.2. Real-Life Test Set Results for Individual Classes*

Table A5 reports the results of the real-life test set referring to the WPS holdout participants. Table A6 reports the same results but referring to the SHS participants.

**Table A5.** Single class results of the baseline and augmented model for the real-life test set. Considered data: WPS holdout participants.

| Mdl | Metric | Stairs Ascent | Stairs Descent | Static | Walking | Wheelchair |
|---|---|---|---|---|---|---|
| **Base.** | **precision** | 0.88 ± 0.02 | 0.98 ± 0.01 | 0.97 ± 0.01 | 0.96 ± 0.01 | 0.23 ± 0.12 |
| | **recall** | 0.87 ± 0.03 | 0.84 ± 0.03 | 0.96 ± 0.0 | 0.79 ± 0.12 | 0.88 ± 0.03 |
| | **f1-score** | 0.87 ± 0.02 | 0.9 ± 0.02 | 0.97 ± 0.0 | 0.86 ± 0.08 | 0.35 ± 0.15 |
| | **specificity** | 0.99 ± 0.0 | 1.0 ± 0.0 | 0.99 ± 0.0 | 0.95 ± 0.0 | 0.88 ± 0.08 |
| **Aug.** | **precision** | 0.94 ± 0.0 | 0.99 ± 0.0 | 0.98 ± 0.0 | 0.95 ± 0.0 | 0.41 ± 0.09 |
| | **recall** | 0.81 ± 0.01 | 0.83 ± 0.01 | 0.98 ± 0.0 | 0.92 ± 0.03 | 0.93 ± 0.0 |
| | **f1-score** | 0.87 ± 0.01 | 0.90 ± 0.01 | 0.98 ± 0.0 | 0.94 ± 0.02 | 0.56 ± 0.09 |
| | **specificity** | 1.0 ± 0.0 | 1.0 ± 0.0 | 0.99 ± 0.0 | 0.93 ± 0.0 | 0.96 ± 0.02 |

**Table A6.** Single class results of the baseline and augmented model for the real-life test set. Considered data: SHS participants.

| Mdl | Metric | Stairs Ascent | Stairs Descent | Static | Walking | Wheelchair |
|---|---|---|---|---|---|---|
| **Base.** | **precision** | 0.80 ± 0.02 | 0.95 ± 0.02 | 0.92 ± 0.01 | 0.97 ± 0.0 | 0.13 ± 0.04 |
| | **recall** | 0.91 ± 0.01 | 0.84 ± 0.02 | 0.93 ± 0.0 | 0.79 ± 0.06 | 0.93 ± 0.01 |
| | **f1-score** | 0.85 ± 0.01 | 0.89 ± 0.02 | 0.92 ± 0.0 | 0.87 ± 0.04 | 0.23 ± 0.07 |
| | **specificity** | 0.99 ± 0.0 | 1.0 ± 0.0 | 0.97 ± 0.0 | 0.95 ± 0.01 | 0.88 ± 0.05 |
| **Aug.** | **precision** | 0.84 ± 0.01 | 0.93 ± 0.0 | 0.91 ± 0.0 | 0.97 ± 0.0 | 0.18 ± 0.03 |
| | **recall** | 0.88 ± 0.01 | 0.83 ± 0.02 | 0.94 ± 0.0 | 0.85 ± 0.02 | 0.93 ± 0.01 |
| | **f1-score** | 0.86 ± 0.01 | 0.88 ± 0.01 | 0.92 ± 0.0 | 0.90 ± 0.01 | 0.30 ± 0.04 |
| | **specificity** | 0.99 ± 0.0 | 1.0 ± 0.0 | 0.97 ± 0.0 | 0.94 ± 0.0 | 0.92 ± 0.02 |

*Appendix A.3. Fully-Rotated Test Set Results for Individual Classes*

Table A7 reports the results for the baseline and augmented models of the fully-rotated test sets referring to the WPS holdout participants. Table A8 reports the same results but referring to the SHS participants.

**Table A7.** Single class results of baseline and augmented model for the fully-rotated test set on every axis. Considered data: WPS holdout participants.

| Mdl | Ax | Metric | Stairs Ascent | Stairs Descent | Static | Walking | Wheelchair |
|---|---|---|---|---|---|---|---|
| **Base.** | X | precision | 0.34 ± 0.32 | 0.40 ± 0.34 | 0.99 ± 0.01 | 0.53 ± 0.23 | 0.43 ± 0.29 |
| | | recall | 0.66 ± 0.23 | 0.56 ± 0.31 | 0.58 ± 0.23 | 0.91 ± 0.06 | 0.33 ± 0.29 |
| | | f1-score | 0.37 ± 0.3 | 0.39 ± 0.3 | 0.70 ± 0.16 | 0.65 ± 0.17 | 0.32 ± 0.27 |
| | | specificity | 0.99 ± 0.02 | 0.97 ± 0.06 | 0.69 ± 0.21 | 0.93 ± 0.05 | 0.92 ± 0.1 |
| | Y | precision | 0.84 ± 0.03 | 0.79 ± 0.05 | 0.95 ± 0.01 | 0.96 ± 0.02 | 0.66 ± 0.17 |
| | | recall | 0.85 ± 0.04 | 0.97 ± 0.01 | 0.95 ± 0.02 | 0.96 ± 0.01 | 0.55 ± 0.2 |
| | | f1-score | 0.84 ± 0.03 | 0.87 ± 0.03 | 0.95 ± 0.01 | 0.96 ± 0.01 | 0.57 ± 0.13 |
| | | specificity | 0.99 ± 0.0 | 1.0 ± 0.0 | 0.99 ± 0.01 | 0.94 ± 0.01 | 0.98 ± 0.02 |
| | Z | precision | 0.43 ± 0.32 | 0.65 ± 0.21 | 0.98 ± 0.02 | 0.57 ± 0.23 | 0.48 ± 0.3 |
| | | recall | 0.61 ± 0.2 | 0.59 ± 0.24 | 0.73 ± 0.26 | 0.95 ± 0.03 | 0.19 ± 0.15 |
| | | f1-score | 0.46 ± 0.28 | 0.58 ± 0.19 | 0.81 ± 0.19 | 0.69 ± 0.18 | 0.23 ± 0.18 |
| | | specificity | 0.98 ± 0.01 | 0.95 ± 0.05 | 0.79 ± 0.26 | 0.95 ± 0.04 | 0.90 ± 0.09 |
| **Aug.** | X | precision | 0.74 ± 0.07 | 0.71 ± 0.13 | 0.98 ± 0.01 | 0.95 ± 0.02 | 0.91 ± 0.04 |
| | | recall | 0.91 ± 0.03 | 0.96 ± 0.03 | 0.96 ± 0.03 | 0.93 ± 0.02 | 0.59 ± 0.17 |
| | | f1-score | 0.82 ± 0.05 | 0.81 ± 0.1 | 0.97 ± 0.01 | 0.94 ± 0.02 | 0.70 ± 0.12 |
| | | specificity | 0.99 ± 0.0 | 1.0 ± 0.0 | 0.99 ± 0.01 | 0.90 ± 0.03 | 0.98 ± 0.02 |
| | Y | precision | 0.75 ± 0.04 | 0.81 ± 0.02 | 0.96 ± 0.0 | 0.97 ± 0.01 | 0.89 ± 0.04 |
| | | recall | 0.92 ± 0.02 | 0.97 ± 0.02 | 0.98 ± 0.0 | 0.94 ± 0.01 | 0.64 ± 0.13 |
| | | f1-score | 0.83 ± 0.03 | 0.88 ± 0.02 | 0.97 ± 0.0 | 0.96 ± 0.01 | 0.74 ± 0.07 |
| | | specificity | 0.99 ± 0.0 | 1.0 ± 0.0 | 0.99 ± 0.01 | 0.94 ± 0.01 | 0.98 ± 0.02 |
| | Z | precision | 0.70 ± 0.08 | 0.79 ± 0.06 | 0.97 ± 0.01 | 0.92 ± 0.03 | 0.93 ± 0.02 |
| | | recall | 0.93 ± 0.02 | 0.96 ± 0.02 | 0.98 ± 0.01 | 0.93 ± 0.02 | 0.42 ± 0.09 |
| | | f1-score | 0.79 ± 0.06 | 0.86 ± 0.04 | 0.97 ± 0.0 | 0.93 ± 0.02 | 0.57 ± 0.08 |
| | | specificity | 0.98 ± 0.01 | 0.95 ± 0.05 | 0.79 ± 0.26 | 0.95 ± 0.04 | 0.90 ± 0.09 |

**Table A8.** Single class results of baseline and augmented model for the fully-rotated test set on every axis. Considered data: SHS participants.

| Mdl | Ax | Metric | Stairs Ascent | Stairs Descent | Static | Walking | Wheelchair |
|---|---|---|---|---|---|---|---|
| **Base.** | X | precision | 0.40 ± 0.30 | 0.40 ± 0.31 | 0.98 ± 0.03 | 0.64 ± 0.13 | 0.40 ± 0.32 |
| | | recall | 0.53 ± 0.24 | 0.49 ± 0.30 | 0.61 ± 0.17 | 0.96 ± 0.03 | 0.17 ± 0.17 |
| | | f1-score | 0.41 ± 0.26 | 0.40 ± 0.28 | 0.73 ± 0.11 | 0.76 ± 0.09 | 0.20 ± 0.18 |
| | | specificity | 0.99 ± 0.02 | 0.99 ± 0.02 | 0.75 ± 0.14 | 0.94 ± 0.04 | 0.94 ± 0.07 |
| | Y | precision | 0.89 ± 0.02 | 0.81 ± 0.02 | 0.91 ± 0.01 | 0.90 ± 0.02 | 0.70 ± 0.24 |
| | | recall | 0.80 ± 0.03 | 0.93 ± 0.02 | 0.90 ± 0.02 | 0.96 ± 0.01 | 0.24 ± 0.09 |
| | | f1-score | 0.84 ± 0.02 | 0.87 ± 0.02 | 0.91 ± 0.01 | 0.93 ± 0.01 | 0.34 ± 0.09 |
| | | specificity | 0.99 ± 0.0 | 1.0 ± 0.0 | 0.97 ± 0.01 | 0.92 ± 0.01 | 0.96 ± 0.02 |
| | Z | precision | 0.44 ± 0.36 | 0.68 ± 0.17 | 0.96 ± 0.04 | 0.63 ± 0.13 | 0.65 ± 0.31 |
| | | recall | 0.56 ± 0.16 | 0.62 ± 0.25 | 0.72 ± 0.21 | 0.98 ± 0.01 | 0.15 ± 0.10 |
| | | f1-score | 0.42 ± 0.28 | 0.61 ± 0.17 | 0.80 ± 0.14 | 0.76 ± 0.09 | 0.18 ± 0.08 |
| | | specificity | 0.99 ± 0.01 | 0.98 ± 0.03 | 0.82 ± 0.19 | 0.97 ± 0.02 | 0.89 ± 0.09 |
| **Aug.** | X | precision | 0.80 ± 0.05 | 0.72 ± 0.11 | 0.94 ± 0.02 | 0.86 ± 0.03 | 0.86 ± 0.08 |
| | | recall | 0.81 ± 0.08 | 0.84 ± 0.07 | 0.88 ± 0.04 | 0.96 ± 0.01 | 0.24 ± 0.10 |
| | | f1-score | 0.80 ± 0.06 | 0.77 ± 0.09 | 0.91 ± 0.02 | 0.91 ± 0.02 | 0.37 ± 0.10 |
| | | specificity | 0.99 ± 0.0 | 1.0 ± 0.0 | 0.95 ± 0.02 | 0.93 ± 0.01 | 0.94 ± 0.03 |
| | Y | precision | 0.84 ± 0.03 | 0.81 ± 0.02 | 0.92 ± 0.01 | 0.90 ± 0.02 | 0.81 ± 0.09 |
| | | recall | 0.88 ± 0.02 | 0.91 ± 0.02 | 0.90 ± 0.01 | 0.96 ± 0.0 | 0.26 ± 0.04 |
| | | f1-score | 0.86 ± 0.02 | 0.86 ± 0.02 | 0.91 ± 0.01 | 0.93 ± 0.01 | 0.38 ± 0.04 |
| | | specificity | 1.0 ± 0.0 | 1.0 ± 0.0 | 0.96 ± 0.0 | 0.92 ± 0.01 | 0.96 ± 0.01 |
| | Z | precision | 0.79 ± 0.05 | 0.80 ± 0.05 | 0.93 ± 0.01 | 0.85 ± 0.02 | 0.92 ± 0.01 |
| | | recall | 0.86 ± 0.04 | 0.89 ± 0.04 | 0.89 ± 0.02 | 0.96 ± 0.0 | 0.19 ± 0.03 |
| | | f1-score | 0.82 ± 0.04 | 0.84 ± 0.03 | 0.91 ± 0.01 | 0.90 ± 0.02 | 0.31 ± 0.04 |
| | | specificity | 1.0 ± 0.0 | 1.0 ± 0.0 | 0.96 ± 0.01 | 0.93 ± 0.01 | 0.93 ± 0.01 |

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
