# Peer review of "Device Orientation Independent Human Activity Recognition Model for Patient Monitoring Based on Triaxial Acceleration"

_applsci, doi:10.3390/app13074175_

Round 1

Reviewer 1 Report

The design of the experiment is reasonable, and the data are detailed. Congratulations on your interesting and practical work. My main concern is expression.

1. Cross-validation: 10:2:3, five-fold. OK. But how was the data separated for CV? Randomly? Orderly? I.e., how to reproduce your experimental outcomes?

2. When describing motion around an axis, such as r1-3, x, y, z is an acceptable but not always uniform description. Try to use the name of the plane to give a clearer explanation, such as sagittal, frontal...

3. When an abbreviation first appears, give the full name. Such as DL

4. There are frequent typos, grammar, and punctuation errors. Such as splitTED; FEWER (instead of LESS) false negatives; low-speed, high-speed; future WORK (no s)... There are dozens of flaws.

5. "In addition to ML, Deep Neural Networks (DNN)-based algorithms have the potential to provide new innovative and efficient solutions, outperforming the ML approaches." You cannot draw this all-inclusive conclusion. HAR is a research topic highly influenced by application scenarios, domains, sensor types, and data corpus, among others, and globally "DNN outperforms ML" is not valid.

A proof from last year from Portugal is https://doi.org/10.3390/s22197324.

Another case: When it comes to the comparison between HMM and NN, the CSL-SHARE dataset (Germany), UniMiB SHAR dataset (Italy), and Enabl3S dataset (United States) were applied. The newly proposed Motion Units (MU)-based HMM (Generalized Sequence Modeling of Human Activities for Sensor-Based Activity Recognition) performed better than or on par with various NN state-of-the-arts. Even if some accuracies were tied, the model of MU-based HMM, drawing on knowledge of automatic speech recognition and kinesiology, is highly simplified, efficient to train, fully INTERPRETABLE, and has great GENERALIZATION CAPABILITY for quick modeling new activities. For example, for the CSL-SHARE dataset, the recognition rate in https://doi.org/10.1016/j.procs.2022.11.220 (Thailand, 2022) applying deep learning (even ResNet) is significantly inferior compared to "Biosignal processing and activity modeling for multimodal human activity recognition" (Germany, 2021) with MU-based HMM, and the NN modeling obviously lost the interpretability, expandability, universality, and generalizability further.

BTW, HMM modeling has also different topologies, such as single-state, fixed numbers of multiple states, adaptive number of states… HMM has also advanced hierarchical models such as HHMM. These research also generated promising outcomes for HAR and even fall detection: Hidden Markov Model and Its Application in Human Activity Recognition and Fall Detection: A Review.

Each sentence in an academic paper should be subject to evidence (literature, experiments, proof, ...). DNN might work better in your task, your approach, or your dataset, but such an arbitrary pervasive conclusion is inadvisable and offensive. Please show respect to peer workpieces, and narrow your statement.

Reviewer 2 Report

This paper focuses on the importance of the device orientation when a three-axial acceleration solution for Human Activity Recognition. Accordingly, the aim of this research was to investigate the impact of changes in sensor orientation on a Deep Learning algorithm targeted at patient-like activities (healthcare assistive tasks oriented).

Two datasets were used. The authors did not explicitly say who acquired this data and how ethical issues were addressed. are sensory data collected by the authors themself? in which contexts? Did Individuals involved are involved in medical treatments or have any specific medical diagnosis? were they aware of the data collection? 

The core of the paper is a DL architecture (standard) fed by augmented data and experimental proofs demonstrated that this helps in reducing the effect of human activity recognition performance. 

No comparisons to other approaches have been proposed. 

summing up, in my opinion, the paper requires substantial improvements before being considered for publication.

Round 2

Reviewer 1 Report

I am OK with the revision.

Reviewer 2 Report

The authors replied to my comments. 

I'm now convinced that the manuscript  deserves publication. 

Congratulations.